# Newspaper Headlines and Intimate Partner Femicide in Portugal

Ariana Correia [1,2,*] and Sofia Neves [1,2]

1   Department of Social and Behavioral Sciences, University of Maia, 4475-690 Maia, Portugal; asneves@umaia.pt
2   Interdisciplinary Centre for Gender Studies (CIEG), Institute of Social and Political Sciences, University of Lisbon (ISCSP, ULisbon), 1300-663 Lisbon, Portugal
*   Correspondence: acorreia@umaia.pt

**Abstract:** The media's representation of intimate partner femicides has been contributing to addressing gender-based violence as a structural phenomenon. Aiming to understand which crime elements are valued and how they might contribute to victim blaming, the present study explores the portrayal of intimate partner femicides in Portugal through the analysis of newspaper headlines. The core of the analysis comprises 853 newspaper headlines published between 2000 and 2017, which were subjected to a categorical content analysis. The results suggest two major trends that are aligned with the scope of the two newspapers analyzed. While some headlines offer informative perspectives on crime and its characteristics, the majority tend to sensationalize the narratives, potentially legitimizing violence against women. The results of this study enrich the social and academic debate on the media's potential influence in preventing and combating gender-based violence. Moreover, by shedding light on the media's representation of intimate partner femicides, the study reinforces the importance of a broader discussion on the role of journalism in fostering social change.

**Keywords:** media; femicide; Portugal; newspapers; qualitative research; gender-based violence

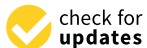



## 1. Introduction

Media have been recognized as sources of information, often the only ones, about crime. The news-making process, from gatekeeping up to framing is highly symbolic, defining the topics on the public agenda, influencing social discourses and practices, and ultimately, contributing to political, social, and educational changes, pointed out as the fourth estate and a vital part of the democratic government. However, the news-making process is guided by news values that determine the newsworthiness of the events, mediated by an individual or a group that, consciously or not, mirrors beliefs, assumptions, and stereotypes together with editorial and organizational guidelines and budget limitations (Carlyle et al. 2008; Silveirinha 2006; Taylor 2009). The media should inform, educate, and encourage social debate, as outlined in the Deontological and Ethical Code of Journalism and emphasized in the Istanbul Convention. When reporting on gender-based violence, sensationalism, which involves spreading fake news to mislead audiences, is prevalent due to uncommited journalism driven by questionable values. Media literacy is essential in preventing sensationalism (Meyers 1997; Ross and Carter 2011; Simões 2008).

According to the UNODC (2022), 45,000 women and girls (56% of total femicides) were killed by their actual or former intimate partners or family, making the home the most dangerous place for women and girls. As widely recognized, media shape the social construction of a crime. Focusing specifically on intimate partner femicide (IPF), the frame used to portray it can promote either education, information, and social engagement, or myths and stereotypes that blame the victims and detach the social and political system from gender-based crimes (Carlyle et al. 2008; Comas-d'Argemir 2014; Fairbairn and Dawson 2013; Taylor 2009).

To frame the present study, which describes the results of qualitative research on the analysis of newspaper headlines of IPFs in Portugal, sections on the mediatic construction of crime and the effects of the mediatization of gendered crime will be presented first.

### 1.1. The Mediatic Construction of Crime

Mass media were developed to disseminate information to a target audience, operating simultaneously as agents of knowledge and as a primary source of information. Under a well-defined criterion, whether sectorial, editorial, and/or organizational, and reflecting individual, social, cultural, and political beliefs, the mediatic discourses (re)construct reality, guiding the public agenda and social discourses (Berger and Luckmann 2004; Lippman 2008).

Hall (1982) suggested that media texts contain a variety of messages and what is presented is just a re-presentation. For that reason, media have the power to signify events in a particular way, unfolding the question: What patterns sustain the construction of the news?

According to Silveirinha (2006), audiences are fueled by tragedy that responds to a voyeuristic interest and emotional appeal. Therefore, newsworthiness, determined from a series of news values that guide and feed the public interest in an event, tends to value negativity, conflict, unusualness, and death, confirming the media saying bad news is good news (Simões 2014).

News values are operationalized through gatekeeping: the selection process that defines which events will make it as public news and which will not. These guidelines, which sustain the news-making process, clear the way for agenda setting and framing (Sydnor 2018).

Framing, described as a "bridging concept between cognition and culture" (Gamson et al. 1992, p. 384) is a powerful way to influence and shape social discourses about crime, comprising the selection and highlight of certain features of a message, held as important, and deemphasizing other features (Goffman 1974). As a communicative process, framing involves frame building (how frames emerge), frame setting (how they reverb individual, social, editorial, and organizational thinking), and frame effects, perceived both on an individual and societal level. On an individual level, news framing may alter beliefs and, consequently, attitudes about an issue. On a societal level, news framing may impact decision making, the political agenda, and collective actions (Scheufele 2006; Vresse 2005), aligned with Entman's (1993, p. 52) concept of framing: "To frame is to select some aspects of a perceived reality and make them more salient in a communicating text, In such a way as to promote a particular problem definition, causal interpretation, moral evaluation, and/or treatment recommendation".

In a seminal study, Iyengar (1991) hypothesized that different types of news frames, concretely episodic and thematic frames, can produce different effects on how the audience perceives and attributes both the causes and solutions of social problems. Within a thematic frame, the focus is on the issue, with the topics and events being presented in a contextualized form, considering a broader and deeper perspective that answers the questions "How and why?", which promotes accountability, social commitment, and mediatic literacy. An episodic frame, on the other hand, focuses on a single event and its protagonists, from a mostly descriptive standing point and answers the questions "What, where, when, and who?", which promotes social problems as limited and niche events (Feezell et al. 2019; Vresse 2005). Iyengar (1991) concluded that news coverage is strongly skewed towards an episodic frame, with the news-making process reinforcing episodic framing, and consequently, promoting stereotypes and oversimplifying social problems.

The episodic frame is the most common in crime news coverage, with a narrative that tends to be descriptive and superficial, prioritizing authorities' sources and forensic information, resorting to technical terminology that is sometimes used in a depthless form. Therefore, the crime tends to be presented in a superficial outlook, frequently exploring the individual characteristics of the protagonists as the central information and motive behind

the crime perpetration, feeding myths and stereotypes related to crimes, their perpetrators, and victims (Chesney-Lind and Chagnon 2017; Wozniak and McCloskey 2010).

In the media, (...) "information is not a passive resource waiting to be searched for and accessed. It is always in competition for people's attention with other information" (Durant and Lambrou 2009, p. 28). Here, headlines have a massive importance since they are the readers' "hook".

According to Bednarek and Caple (2012), headlines have multiple functions, like an informative function for summarizing the story; a framing function, providing a position towards the event; and a news value function, maximizing the newsworthiness. Still, headlines as precise news depictions are decreasing, with other priorities rising, such as getting the reader's attention and promoting biased representations (Andrew 2007).

Several studies have highlighted the role of the headline style in directing readers' information processing (framing) or inferencing, facilitating the social construction of the crime (Bednarek and Caple 2012; Molek-Kozakowska 2013; Van Dijk 2003). When the crime is gender-based, the news coverage can reproduce and legitimize conservative gender beliefs, which promote social apathy regarding gender equality and social justice, in a news-making process that is never neutral but instead is socially constructed (Baldry and Pagliaro 2014; Belknap 2007; Comas-d'Argemir 2014; Fairbairn and Dawson 2013).

*1.2. The Mediatic Discourses on Gendered Crime*

The media coverage of gender-based crimes, with femicides being the lethal outcome of this phenomenon, can play a powerful role in addressing gender-based violence as a social problem, promoting audience engagement, which can lead to institutional and governmental changes. Contrarily, mediatic discourses that promote victim-blaming narratives, which minimize or normalize gender-based violence, lead to perpetrators' impunity, creating obstacles for women's rights (Gillespie et al. 2013; Richards et al. 2011; Taylor 2009). In this sense, media coverage of gender-based crimes mirrors a society's stance regarding gender asymmetries (Comas-d'Argemir 2014; Meyers 1997).

Beyond gender, the sexual orientation, gender identity, socioeconomic status, cultural background, or religion of the criminal news protagonists are not neutral in terms of their mediatic value, especially when these categories intersect, facilitating a blaming of the victim narrative, which is extremely relevant material to an intersectional approach (Crenshaw 2002; Smiley and Fakunle 2016).

Berns (2004), in a providential work on victim framing in the media, identified three approaches to intimate gender-based violence on news: (i) the empowerment perspective, which focuses on the victim's resilience, in an inspiring and learning approach; (ii) the social justice perspective, that guides attention to the social and cultural matrices of gender-based violence; and (iii) the victim-blaming approach, with a narrative built around the victim's accountability, underpinning individual traits and relational dynamics, shoving intimate violence to the private sphere. This victim-blaming portrait is even more evident when the victims (women) do not fulfill gender roles, resulting in them being presented as *less victims* and therefore, less needed, contrasting with the ideal of the victim, where women are portrayed as passive, sufferer, submissive, caring, and kind (Capezza and Arriaga 2008; Lelaurain et al. 2017).

In a systematic review of media representations of violence against women from 16 countries, particularly printed newspapers, Sutherland et al. (2016) underlined several news coverage characteristics, which are very enlightening: (i) the absence of the reference of a social context; (ii) the resource to sensationalist strategies, namely the disproportional attention brought to the protagonists and the event itself; (iii) the perpetuation of myths and stereotypes regarding gender-based violence, skewing public opinions on the matter; (iv) the prioritization of authorities as the primary information sources instead of specialists and researchers (if there is no previous report, it does not mean there was no previous victimization); and (v) the resource to direct and indirect victim blaming.

When the focus is on IPF, with the femicide being perpetrated by a former or actual partner, these news coverage features can be amplified (Comas-d'Argemir 2014).

### 1.3. The Mediatic Discourses on Intimate Partner Femicide

Russell recovered the term femicide in 1976, conceptualizing it as "the killing of women because they are women" (Russell and Harmes 2001, p. 13), sustaining that the neutrality and broadness of the term homicide leave out the political grid of this crime, making it invisible and easier to bias. Later, Caputi and Russell (1992) extended the definition by considering femicide as the most extreme outcome of a *continuum* of gender-based violence. Therefore, it must not be portrayed as a single, impulsive event or a crime of passion, but the final power and control (Campos 2015).

According to the UNODC (2022), per hour, on average, more than five women or girls are murdered worldwide due to gender-related motives. Because 4/10 femicides do not have enough information attached, it is surely an underreported crime disclosing a dark reality, that being home is the most dangerous place for women and girls.

Media have been pointed out as key platforms in preventing and combating violence against women in its various forms. The Istanbul Convention (European Comission 2011, in its 17° article—Participation of the private sector and the media—addresses media as essential in changing the attitudes of the public, overcoming gender stereotypes, and raising awareness of the various forms of violence against women. However, an independent expert body responsible for monitoring the implementation of the Istanbul Convention— GREVIO (2019)—pointed out several challenges in applying the amendments, namely the development and monitorization of self-regulatory standards and ethical codes for these topics, as well as the lack of gender-based violence training in media professionals.

Although it has not been explored much in academia, the studies on the media coverage of IPFs have highlighted victim-blaming features through direct and indirect tactics. The direct victim-blaming tactic is perceived as a negative overall description of the victims, laying emphasis and/or speculating about her habits, her personal, familiar, and professional history as triggers to the femicide and ignoring contextual factors, namely prior victimization and relational dynamics, as well as social factors such as living in a community that legitimizes intimate partner violence and discourages its report to authorities. The indirect victim-blaming tactic minimizes the responsibility of the perpetrator, often through psychopathology assumptions, personality traits, intergenerationally violence, and/or the consumption of alcohol and drugs (Meyers 1994; Bullock and Cubert 2002; Gillespie et al. 2013; Richards et al. 2011; Taylor 2009; Simões 2008; Wozniak and McCloskey 2010). Here, IPF is often portrayed as a crime of passion, with jealousy, humiliation, and rejection at its core, which activate biased information (Neves 2016; Simões 2008).

Overall, IPF coverage is superficial and unprecise, being that the narratives are frequently dramatized and simplified within a sensationalist approach, making this news more appealing to the consumers, recentering the media coverage on editorial and budget pressures (ultimately, profit matters), with severe potential impacts on social discourses and practices (Molek-Kozakowska 2013).

### 1.4. From Social Discourses to Practices: The Effects of the Mediatization of Gendered Crime

The literature is consistent in recognizing the potential effects of mediatic narratives, namely the biased social construction of crime and the social engagement, or not, with the issue (Carlyle et al. 2008).

#### 1.4.1. The Social Construction of Crime

Here, framing plays a major role, with more frequent ones directly linked to the social acceptance of the given portrait of crime and its protagonists. Nonetheless, not all frames present the same potential to affect the audience, only the ones that resonate with the consumers. Resonance, which is related to the frame appeal, making it appear natural and familiar, was highlighted as a key factor for media influence by Gamson et al.

(1992). Later, Benford and Snow (2000) broke down resonance into two criteria: credibility and salience. While credibility relates to information consistency, i.e., not contradicting previous information, salience is perceived when the used frame is aligned with the previous experiences of the consumers, increasing its influence potential.

Besides resonance, the media coverage's potential to influence the social construction of crime increases when fear underlines the narrative. Particularly vulnerable media consumers (as intimate partner victims) can be easily influenced by media portrayals of the crime, as well as consumers who share social and demographic features with the protagonists. Also, when the audience lacks information about the crime, they become more receptive to the portrait presented (Carlyle et al. 2008; Surette 2014). Then, it presents three of the most referred potential impacts of media coverage on IPF.

### 1.4.2. Copycat Crime

The copycat effect is defined as a criminal act that is modeled or inspired by a previous crime that has been reported in the media or described in fiction (Helfgott 2015). This effect has been related to suicide, named the Werther Effect, after the eighteenth-century novel The Sorrows of Young Werther written by Johann Wolfgang von Goethe, where the central character commits suicide, and shortly after the book launched, a series of identical suicides are registered, in a supposed mimetic effect (Niederkrotenthaler et al. 2010). Similarly, some studies have hypothesized that news coverage on IPF acts as a trigger for the perpetration of an identical crime, after the identification of the perpetrator with the mediatized crime (e.g., Vives-Cases et al. 2009; Torrecilla et al. 2019).

### 1.4.3. Disempowerment

IPFs seem to affect the well beings of intimate partner violence victims, as they are at a higher risk of a lethal outcome (Campos 2015). News coverage of the crime can increase their fear, social isolation, and perceived insecurity, as well as promote disbelief in justice and discourage reporting to authorities and specialized support, leaving these women particularly vulnerable and unprotected (Carlyle et al. 2008; Comas-d'Argemir 2014).

### 1.4.4. The Protective Effect

News coverage, when focusing on initiatives designed to prevent and/or intervene in intimate partner violence or femicide, such as legal measures, public policies, and educational actions, seems to dissuade intimate partner perpetrators (Vives-Cases et al. 2009; Carlyle et al. 2014).

## 2. Materials and Methods

The present study is part of a broader research topic. It aimed to analyze news headlines on IPF crimes to understand which crime elements are valued and determine if they promote victim blaming. In doing so, a specific goal can be achieved: to comprehend the connections between the most valuable criminal elements of the cases and their associations with the direct or indirect blaming of victims.

By analyzing news headlines on IPF, awareness about the issue can be raised, bringing to attention its complexity and severity. Moreover, understanding the social narratives constructed by the media can be insightful for deconstructing victim blaming and stereotypes.

### 2.1. Theoretical Framework and Research Method

The approach that underlies this study is intersectional feminism. Thus, the concept of intersectionality, brought to light by Kimberly Crenshaw in 1989, applies to this research, since femicide is perceived as the most extreme form of gender-based violence, making a broader analysis through the social, cultural, and political lenses of this phenomenon mandatory (Crenshaw 2002; Russell and Harmes 2001). At the same time, media are highly recognized as sources of social influence, operating both as agents of knowledge and primary sources of information, reverb social beliefs, editorial guidelines, and organizational

priorities, transforming news into a symbolic social, cultural, and political construction rather than a neutral frame of an event (Berger and Luckmann 2004; Lippman 2008). This study is also aligned with Fairclough's theory of language and power, which perceives language as a social and powered practice (Fairclough 1992).

*2.2. Core of Analysis*

The cote of the analysis comprises the headlines of 853 news articles published over 18 years, between 2000 and 2017. The selected period was chosen since domestic violence became a public crime in 2000, meaning that the legal process no longer required the consent of the victims. This change in the legal nature of the crime has constituted a milestone in addressing domestic violence as a social problem. Additionally, 2017 was the previous year before the implementation of the National Strategy for Equality and Non-Discrimination 2018–2030, which marks an innovative approach regarding women's rights, aligned with the UN 2030 Agenda.

The news pieces were retrieved from printed editions of two Portuguese journals with totally different targets and characteristics. One, Correio da Manhã (CM), the best-selling journal in the country, has a sensationalist editorial guideline, prioritizes short and descriptive news stories, has appealing headlines, and explores non-central features of the crime. The other journal, Público (P), is characterized by its robust informative scope, prioritizing bigger dimensions of a news story to contextualize the crime (Carvalho 2007).

From the total, 637 (74.7%) of the news stories were published in CM and 216 (25.3%) in P.

*2.3. Procedure*

Authorization from the municipal libraries of Porto and Braga was acquired to consult the newspapers in person and collect the data. The process lasted over one year to be completed, with weekly visits. Once pre-analyzed, the news that met the inclusion criteria, namely news regarding intimate femicides consummated or attempted, perpetrated between 2000 and 2017, and published in CM or P, were photographed and stored. Later, the contents were compiled in a database using SPSS software, version 26.

*2.4. Data Treatment and Analysis*

All the news stories were registered using the initials of the title of each newspaper, followed by the publication year and month and a number (e.g., CM2002_10_01; P2002_10_01). To include all relevant variables, such as the victim and offender characteristics, motives, societal reactions, and legal aspects, a coding scheme was developed. Afterward, the news story was subjected to a coding process in terms of the format and content. In an early phase, two researchers codified the data autonomously. A categorical content analysis (Bardin 2011) was adopted to identify the most valuable criminal elements of the cases.

To reach an intercoder consensus, regular meetings were organized. Disagreements and inconsistencies were resolved with the presence of a third-party senior expert, and a final codification was defined, ensuring the reliability and validity of the data analysis.

**3. Results**

In total, 853 news pieces were collected, and correspondingly, 853 headlines on IPF. Among these, a total of 637 (74.7%) were retrieved from CM and 216 (25.3%) from the P newspaper. Of all 637 headlines retrieved from CM, 380 (59.7%) concerned consummated crimes and 257 (40.3%) attempted ones. Of all P (n = 216) news pieces, 165 (76.4%) reported consummated intimate partner femicides and 51 (23.6%) attempted ones.

From all news pieces collected (n = 853), 644 different cases of IPF were identified, with 59.5% (383) successful and 40.5% (261) attempted. A total of 209 of the crimes were reported by both newspapers, 428 were reported only by CM, and 7 crimes were reported only by P.

From an exhaustive reading of the main headlines, seven mediatic narratives emerged (see Table 1): (1) property; (2) victim blaming; (2.1) direct victim blaming and (2.2) indirect victim blaming, where the crimes are justified by jealousy, honor, compassion, or despair; and (3) victim dehumanization, which alludes to a corporate crime and the *modus operandi* pland invisibility of the victim. The crime as an announced tragedy (4) and as a purely technical and formal police-style description (5) also emerged. With a focus on the crime sanctions (6), three discourses stand out: (6.1) the perpetrator's arrest, (6.2) the perpetrator's conviction, and (6.3) legal literacy. Lastly, in terms of the (7) crime particularities, some of the headlines focused on particularities of the crime such as (7.1) the victim's pregnancy, (7.2) the perpetrator's suicide, and (7.3) child witnesses.

**Table 1.** Analysis of intimate partner femicide news.

| Categories | Subcategories |
|---|---|
| Property | |
| Victim blaming | Direct victim blaming<br>Indirect victim blaming |
| Victim dehumanization | The corporate crime<br>The *modus operandi* and invisibility of the victim |
| The announced tragedy | |
| The police style description | |
| The sanction as the headline core | The perpetrator's arrest<br>The perpetrator's conviction<br>Legal literacy |
| Crime particularities | The victim's pregnancy<br>The perpetrator's suicide<br>Child witnesses |

A sub-headline, which contextualizes the main headline, giving complementary information, was present in less than half of the news pieces (40.5%), 28% in CM, and 70% in P. Because the sub-headline works as an extension of the main one, the categories are shared except for the subcategory A crime of despair, a form of indirect victim blaming, which emerged only in the sub-headlines.

The six categories and subcategories that emerged within the categorical content analysis are now presented.

### 3.1. Property

In this category, the headline appeals to direct quotes from the perpetrator, claimed during or before the crime as a threat, frequently during the couples' separation, revealing a discourse of power and control from the perpetrator who perceived the victim as his property, symptomatic of the romanticized belief *If not mine, no one else's*. Inclusively, this sense of property is directly linked with jealousy and coercive behavior that tends to escalate as a demonstration of power (from the perpetrator), being a documented risk factor for intimate partner femicide (Aldridge and Browne 2003; Belfrage and Rying 2004). The quotes are shared by witnesses who were heard, mainly by the CM newspaper. Next, the headlines more representative of this category (our translation) will be presented.

*I am going to take you with me* [CM2010_09_25]

*I killed her, but I still love my wife* [CM2014_11_52]

### 3.2. Victim Blaming

The victim blaming discourse present in both types of headlines, main and sub-headline, centers its narrative on the victim's accountability for the crime, underpinning individual traits and relational dynamics. Therefore, the crime attribution focuses on the

private sphere, totally distant from a sociopolitical context. The victim-blaming discourse was forked regarding its direct or indirect character, which is developed below.

### 3.2.1. Direct Victim Blaming

When the victim-blaming discourse is direct, the crime, either consummated or attempted, stands out and involves the use of a negative language towards the victim, with crime attributions linked to the victim's behavior, frequently futile and deeply associated with conservative gender beliefs. Therefore, these discourses promote a biased understanding of the crime and its social and cultural matrices, with some of the most representative examples presented below.

*Shot the wife because she couldn't cook* [CM2007_12_18]

*Attacked his wife with a knife because she refused sex* [CM2010_04_09]

### 3.2.2. Indirect Victim Blaming

Despite the presence of a victim-blaming narrative, in this category, it appears under an indirect form, focusing on perpetrator complaints that are presented as crime motives, which promotes the reasons for the crime perpetration. This category emerges from a discourse that understands and legitimizes the crime within a condescending narrative that moderates its severity. The indirect victim blaming was reflected in multiple forms, all excusing the crime by the perpetrator's deeply felt emotions, as all were perceived as provoked by the victim, whether the reason was jealousy, despair, compassion, or an honor attack. Therefore, within an indirect victim-blaming narrative, the perpetrator's nonaccountability stands out. Here, the headlines announce morbid jealousy as the crime motive, amplifying the Manichaean, romanticized, and dangerous logic of jealousy, that floats from being an indicator of a healthy relationship to being pathologically driven. The association of jealousy with love, which understands impulsive, erratic, and mad behavior, contributes to framing IPF as an unpredictable and passionate crime as in these examples: "Set his wife on fire because of mad jealousy" [CM2000_06_12]; "Kills his wife and two daughters out of madness" [CM2007_12_17]. The (perceived) infidelity of the victim, which, irrespective of its veracity, functions as the crime motive and places the honor of the perpetrator at the center of the crime, was one of the most recurring themes in the news. Simultaneously, the motive underlined brings a promiscuous aura to the deceased woman and, in a certain way, as deserving of her fate, as evident in the following headlines: "Stabbed for love. Ran away with another man. Two-year-old son saw everything" [CM2007_10_14]; "Kills ex-wife with a shotgun and injured his rival. Waited for the ex-wife and her boyfriend on their property and shot them" [CM2008_10_30].

Mercy as a motive for the femicide was also perceived within an indirect victim-blaming narrative, where the crime was dissociated from an intimate partner violence situation yet emerged from a discourse that framed the crime as euthanasia. Accordingly, the crime is presented as an exercise of love and devotion by the perpetrator to the victim, who is pictured as debilitated yet with a lack of medical proof, as in the following examples: "Kills his wife to end the illness. The couple is found dead. Husband took his wife's life and then killed himself" [CM2008_07_14]; "He was a good man and looked up to her. She had a brain tumor, and he couldn't live without her" [CM2015_04_09]. In the same category, and present only in sub-headlines, is the crime attribution to the perpetrator's despair, frequently post-breakup/divorce, consummated, or intended. Because the separation is the initiative of the victim, the accountability of the crime shifts towards her, and the perpetrator is presented as a suffering man, whose erratic behavior is out of love. The women are frequently presented as cold and not engaged in the relationship, with some of the most representative examples presented next: "Beheaded wife in front of their daughter. Woman was stalked by an ex-husband that never forgave her for the divorce" [CM2005_11_29]; "I killed my wife and son. Act of despair leads Daniel to stab his ex-wife and son" [CM2010_04_07].

### 3.3. Victim Dehumanization

One of the most common narratives presented in the main headlines, not so much in the sub-headlines, was the focus on the perpetrator's status and background, giving disproportionate information regarding the victim, remitting her to a place of no interest, resembling an object. Here, the social and professional backgrounds of the perpetrator and the *modus operandi* description control the narrative, emerging as subcategories, as presented below.

### 3.3.1. The Corporate Crime

The main headlines that introduce the perpetrator through his professional occupation commonly do not present information from the victim. In these cases, the perpetrators' professional occupation is socially valued (e.g., athlete, lawyer, or engineer), blurring the upcoming story. When the victim is introduced through her professional occupation, it is generally one that is not socially valued, if not stigmatized (e.g., sexual worker), hierarchizing the protagonist's social power.

One piece of information that stood out was the number of perpetrators that were in law enforcement. Although the information available does not allow for an association between high-power professional status and gender-based crimes, especially femicide, the link should be investigated.

The information available in these headlines is brief and superficial, with few sub-headlines. Some of the most representative examples are presented below.

*Former criminal police officer shoots dead ex-wife* [CM2011_06_13]

*Former military blows up wife* [CM2015_12_44]

### 3.3.2. The Modus Operandi and Invisibility of the Victim

One of the most frequent headline formulae is the focus on the *modus operandi*, being how the crime was perpetrated, giving a sub-headline attention to the context in which the crime was committed (e.g., domestic violence). Therefore, the victim is present in the narrative through action verbs, sometimes close to animalistic forms like shot, thrown, or *cut* with a graphical semantic construction that appeals to the reader's reaction. This subcategory was significantly present in the CM newspaper, and some of the most representative headlines are presented below.

*Kills ex-girlfriend and throws her to the garbage* [CM2005_09_26]

*Killed his wife with a chainsaw* [CM2010_05_13]

### 3.4. The Announced Tragedy

Both in the main and sub-headings, testimonials from family, friends, and community regarding the crime are frequently presented. Mainly in the CM newspaper, direct quotes are highlighted, most commonly post-crime, for the discourses are emotional and spontaneous. From these reports, a pattern of social inactivity is perceived, once the history of domestic violence is known. In some situations, neighbors reported that they supported the victim in some protection strategies, like letting her spend the night, having a spare key, or being watchful. However, the report to the authorities in these cases was residual, even though domestic violence is a public crime, meaning it can be reported by anyone and anonymously. In some situations that were formally reported to the authorities, several flaws were pointed out, since the criminal classification given (e.g., physical offenses instead of domestic violence which activate different protocols), lack of legal protection (e.g., non-efficient protective measures), and unsuccessful court outcomes mark a fragile system that does not respond to the victims' needs, leading to their deaths.

These headlines were collected mostly from CM, and some of the most representative headlines are presented below.

*Murdered his wife and ran away with their children. He said that one day we would find her dead* [CM2002_05_21]

*Woman death was "announced" six months ago* [CM2007_03_01]

### 3.5. The Police-Style Description

The headlines within this category resorted to police announcements, frequently with technical language (e.g., suspect, allegedly) and being strictly factual and shorter than other headlines, providing no information regarding the social and cultural contexts of the crime and limiting any association between the IPF and intimate partner violence to the existence of a prior formal complaint. In this regard, the lack of a formal complaint of intimate partner violence makes this (strong) association an invisible one. Nevertheless, in the past decade, the use of this type of format has been decreasing, giving way to more informative headlines.

These headlines were collected mostly from P, and some of the most representative headlines are presented below.

*Man is suspect of shooting his wife* [P2002_02_13]

*A man was arrested suspected of killing wife* [P2008_08_03]

### 3.6. The Sanction as a Headline Core

The present category was not as common as the previous ones and was mostly collected from the P newspaper. Here, the focus is not the crime perpetration per se or the protagonists, but the legal consequences of it. This category has been more regularly present in the last years of the analysis, promoting the social, legal, and political visibility of IPFs.

From this category, three discourses emerged, both in the main and sub-headlines, which are developed below.

#### 3.6.1. The Perpetrator's Arrest

The perpetrator's arrest has a central information piece that highlights the authorities' interventions, promoting a sense of protection and justice. Presenting an IPF through a sanctioning frame reinforces the authority's tenacity and the system's health.

*Killer husband goes to jail* [CM2011_08_18]

*Police arrest suspect of wife's death* [P2006_11_04]

#### 3.6.2. The Perpetrator Conviction

Despite the arrest, the prosecution is not guaranteed; therefore, the conviction stands as the ultimate part of the prosecution, with justice pending on the sentence. However, with the focus on perpetrator conviction, there is a sense of an oiled system, worthy of credit.

*Man, who killed wife sentenced to 18 years in jail* [P2002_02_04]

*Heavy sentence for man who killed his wife* [P2014_11_22]

#### 3.6.3. Legal Literacy

An informative discourse highlights the sanction characteristic of the headlines, emerging both in main and sub-heading.

In some cases, these headlines are present in central pieces of the newspaper, like journalistic investigations or regarding official reports or data (e.g., number of domestic violence annual reports), or, in fewer cases, regarding a report on several IPF crimes that took place in a short period. Some examples are presented below.

*At least 22 women were killed by their partners this year. This weekend two women were killed.* [P2013_07_05]

*Two women died at the hands of their partner this weekend. On average 60 women were killed, and 300 children get orphaned due to intimate partner homicide* [P2004_08_05]

### 3.7. Crime Particularities

This last category regards the particularities of some of the crimes, in their consummated or attempted form, that are highlighted in headlines. Thus, three situations emerged, which are presented below with the most representative headline examples.

#### 3.7.1. The Victim's Pregnancy

If the victim was pregnant, it was for sure highlighted in the headline due to the severity of the crime, nevertheless supported by sensationalist strategies like crime details through graphic and emotionally appealing descriptions.

*Eight months pregnant killed by her husband* [P2013_03_13]

*Five months pregnant was dragged by the hair maternity by her husband, who was arrested and released shortly after by the judge* [CM2014_06_32]

#### 3.7.2. Femicide–Suicide

From all the cases of femicide collected (n = 644), 18.9% (n = 122), reported a femicide–suicide case, and suicide attempts were registered in 9.1% (n = 59) of the crimes. Considering only consummated femicide and successful suicide cases, there were 103 cases with fatal outcomes for both, and in 19 cases of suicide, the victim survived.

Regarding the circumstances of the consummated suicide–femicide, in 62.3% (n = 76) of the cases, the couple was together, and in 37.7% (n = 46) of the cases, the couple was separated.

*Murdered his partner and committed suicide* [CM2001_05_08]

*Killed his wife and a friend of hers then committed suicide in front of his daughter's kindergarten* [P2013_01_04]

#### 3.7.3. Child Witnesses

The central information of these headlines is the fact that the crime, whether committed or not, was witnessed by children, normally descendants of both the victim and perpetrator.

*A three-year girls saw her father stab her mother and all because of soup that was undone.* [CM2012_09_31]

*The killer's brother finds four-year-old nephew next to the mother's bloody corpse. The child was taken by the health assistance and handed over to his grandparents* [CM2015_03_01]

## 4. Discussion

Through the newspaper headline analysis, two scopes emerged—informative and sensationalist—with the last one standing out. The sensationalist scope, present in the CM newspaper, promotes a victim-blaming narrative that can be promoted in obvious (i.e., direct) or more subtle (i.e., indirect) ways, both in the main and sub-headlines. In the first one, the crime is justified by the victim's behavior, frequently futile and deeply linked to conservative gender beliefs. Within the indirect forms of victim blaming that emerged from the newspaper headline analysis, the crime stood out as a narrative that understands and, to some extent, justifies the crime perpetration due to the perpetrator's strong emotions, which were *always* provoked by the victim. Thus, an indirect victim-blaming narrative emphasizes intimate partner femicide as an unpredictable crime, committed out of *love*, dissociating the crime from its social and cultural matrices. When framing it as a passionate crime, a clear association between jealousy and love is made, romanticizing impulsive, erratic, and *mad* behavior. A femicide committed as an honor crime, usually related to infidelity suspicion or the non-acceptance of the breakup, brings a promiscuous aura to the deceased woman and, in a certain way, as deserving of her fate. Less frequently, some crimes were framed as euthanasia followed by the perpetrator's suicide, especially in elderly couples, where even though there was a great lack of medical information, the

crimes were framed as mercy killings, highlighting the perpetrator's love for the victim (Richards et al. 2011). Although these categories emerged independently of the victim's age, the crimes framed as mercy crimes and categorized as indirect victim blaming were exclusive of older couples.

The elements that emerged through the headline analysis are consistent with blaming the victims' tactics, whether jealousy, despair, compassion, or an honor attack, previously referred to by several authors, underlining the legitimation of the crime motive, and therefore, the perpetrator's severely diminished accountability (Aldrete and Fernández-Ardèvol 2023; Berns 2004; Bjornstrom et al. 2010; Comas-d'Argemir 2014; Gillespie et al. 2013; Lelaurain et al. 2017; Richards et al. 2011; Smiley and Fakunle 2016; Sutherland et al. 2016; Taylor 2009).

Within a sensationalist scope, the headlines often resorted to the perpetrators' direct quotes through community sources. These quotes frequently slip into a blaming-the-victim narrative, through which the perpetrators' accountability is diminished or even put aside. The category—*The announced tragedy*—present in the main and sub-headings shows a permissive community that is complacent with the crime, since a history of domestic violence was frequently known, remitting it to the private sphere as an excuse for their inaction. Yet, in some cases, the neighbors reported that they supported the victim in some protection strategies (e.g., letting her spend the night, having a spare key, or being watchful), but a formal complaint was not an option, even though it has been a public crime in Portugal since the year 2000, which means it can be/should be reported by any witnesses. This social discourse highlighted in the headline, mainly in the CM newspaper, can constrain actual domestic violence victims to report the crime and find support (Aldrete and Fernández-Ardèvol 2023; Neves 2016).

In several cases, the headlines focused on accessory information, often being crime details in an emotional appeal, remitting the victim to a place of no interest, resembling an object, as a dehumanizing exercise, which sustains the following categories: (i) the *modus operandi* and invisibility of the victim category and (ii) the *modus operandi*, where the perpetrators' professional occupation is socially valued (e.g., athlete, lawyer, or engineer) and when the victim's profession is shared, it is generally not a socially valued one, if not stigmatized (e.g., sexual worker). These categories are aligned with Sutherland et al.'s (2016) systematic review of media representations of violence against women, which underlined the disproportionate attention brought to the protagonists and crime details through graphic and emotionally appealing descriptions as a commonly used sensationalist strategy (Enne 2007).

Almost exclusively in the P newspaper, the headlines are presented using police information, bringing short, factual, and non-contextualized information using terms like suspect or alleged. The overuse of police reports as a primary source of information can lead to a biased portrait of the crime, especially because it is an underreported one (Budó 2016).

The informative scope is present in the category of The sanction as headline core, in which the P newspaper stood out, focusing on the criminal consequences and giving visibility to the system of justice, either through the perpetrators' arrest or his conviction. In some cases, the headlines are present in central pieces of the newspaper, like journalistic investigation or regarding official reports or data (e.g., number of domestic violence annual reports), or, in fewer cases, regarding a report on several intimate partner femicides in a short period in a perceived example of legal literacy.

Now, the present study results presented a solid pattern of victim blaming in news headlines of IPF, which is consistent with several studies on this matter, mainly focusing on the news piece contents (Berns 2004; Gillespie et al. 2013; Lelaurain et al. 2017; Richards et al. 2011; Taylor 2009). However, with headlines working as the readers' "hook" to the news, maximizing the event's newsworthiness, their construction is commonly aligned with the news piece, providing a position towards the event, which can be informative or biased (Bednarek and Caple 2012; Molek-Kozakowska 2013).

As crime elements highlighted by the newspapers, the motive, the protagonists' relationship status, the *modus operandi*, and the social reactions to the crime were prominent, frequently exploring satellite factors, coherent with a sensationalist frame, other than its gender-based matrix (Belknap 2007; Carlyle et al. 2008; Fairbairn and Dawson 2013; Richards et al. 2011).

The headline analysis of these two Portuguese newspapers revealed a polarized news construction, which has been more visible since 2009. Even though it is not possible to confirm a connection, this year, Law No. 112 from 16/09 was passed, bringing a special juridical regime to domestic violence victims and bringing more social awareness to the topic. From 2010 on, P stands out for its informative scope and CM maintains its simplistic yet appealing frames. As the most-read newspaper in the country, attention is drawn to the potential impact of the headlines (e.g., mimetic potential: copycat crime, constraining new reports, and fear), as consequently, the news pieces that promote intimate partner crime as a passionate crime are hence impulsive and unpredictable. Therefore, the IPF headlines present the crime as an isolated event, legitimizing and sometimes empathizing with perpetrators' motives that most commonly blame the victim.

## 5. Conclusions

The perception, selection, and transformation of events into news pieces reflect a constructed process, permeable to individual, social, cultural, and political beliefs as well as editorial, economical, ethical, and deontological guidelines, transforming news pieces in highly symbolic products (Berger and Luckmann 2004; Correia et al. 2017).

However, media are not neutral broadcast channels, nor are they consumed passively by the audience, which retrofeeds the mediatic agenda. Although it is not a unidirectional process, the mediatic consumption lacks literacy, becoming fertile soil for media to produce the story while reproducing the dominant cultural order (Hall 1982), with a high impact on the social construction of the crime, especially gender-based crimes like IPFs (Berns 2004; Comas-d'Argemir 2014; Schnepf and Christmann 2023). Also, it urges to address the already perceived potential of IPF as a media-influenced copycat crime, meaning the mediatization of femicide can promote and mimic similar behaviors (Vives-Cases et al. 2009; Helfgott 2015; Surette 2014).

Within news construction ruled by economic guidelines that disregard the main principles of journalism ethics, together with acritical audiences and no robust and effective media monitoring systems, social changes will be indefinitely delayed (Berger and Luckmann 2004).

The present study highlights the importance of framing the news about IPF in Portugal in the sociopolitical and cultural contexts of where they are produced. The sensationalist scope of the media coverage has the potential to reinforce victim-blaming perspectives, and as a consequence, further polarize public opinions in a distortive way.

Aware of the potential negative effects of the media coverage of domestic violence and femicide, the Portuguese government launched, in 2019, a guide for best practices for the media in preventing and combating violence against women and domestic violence (República Portuguesa 2019). Among other goals, the document recommends implementing a problem-oriented strategy to address violence against women in both public and private settings by utilizing tailored concepts and clear, evidence-based language to prevent the potential of encouraging imitation among perpetrators. Also, it is recommended to avoid presenting informative content that justifies, excuses, or establishes causal relationships by highlighting the victim's personality traits, behaviors, or sociocultural background as well as those of the aggressor, to not accentuate the insecurity and vulnerability of the victims. These recommendations are aligned with the Istanbul Convention (European Commission 2011), which urges the media to help create and implement policies, set guidelines, and uphold self-regulatory standards to prevent violence against women and promote respect for their dignity.

**Author Contributions:** Conceptualization, A.C. and S.N.; methodology, A.C.; formal analysis, A.C. and S.N.; writing—original draft preparation, A.C.; writing—review and editing, A.C. and S.N.; funding acquisition, A.C. All authors have read and agreed to the published version of the manuscript.

**Funding:** Fundação para a Ciência e Tecnologia: SFRH/BD/131440/2017.

**Institutional Review Board Statement:** Not applicable.

**Informed Consent Statement:** Not applicable.

**Data Availability Statement:** The raw data supporting the conclusions of this article will be made available by the authors on request.

**Conflicts of Interest:** The authors declare no conflicts of interest.

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
