# Peer review of "Newspaper Headlines and Intimate Partner Femicide in Portugal"

_socsci, doi:10.3390/socsci13030151_

Round 1

Reviewer 1 Report

Comments and Suggestions for Authors

This reviewer thinks the topic and approach is very relevant and timely. The feminist lens used to analyze the selected newspaper headlines is novel and sheds important light on the influence of the media in depersonalizing domestic violence femicides in the interests of attracting readers and not challenging toxic perceptions of women murdered by domestic partners. I woudl like to have seen some discussion of age as an intervening variable.For example, do headlines of reports on femicides involving older women reflect the same depersonalizing language as with younger women? The introduction and discussion of the lens used to analyze relevant headines was excellent. The English used in discussing the actual study was awkward, however, and should be reviewed, however. As an example, lines 272-3 "Lately a registration grid was completed recurring to the data base...." doesn't make much sense ,or 351-2 "the perceived infidelity from the victim that independently from its veracity serves as crime motive.." is unclear. While these lapses are understandable when English is a second language, good proofreading would help smooth this out.

Comments on the Quality of English Language

Please see note to authors

Author Response

Thank you for reviewing our manuscript. The suggestions were considered and included. English was revised. Please check the new version with highlights.

Authors Response:

This reviewer thinks the topic and approach is very relevant and timely. The feminist lens used to analyze the selected newspaper headlines is novel and sheds important light on the influence of the media in depersonalizing domestic violence femicides in the interests of attracting readers and not challenging toxic perceptions of women murdered by domestic partners.

Response: We appreciate your general comments about the manuscript.

I would like to have seen some discussion of age as an intervening variable. For example, do headlines of reports on femicides involving older women reflect the same depersonalizing language as with younger women?

Response: A quote on age was included in the text (see 539-541).

The introduction and discussion of the lens used to analyze relevant headines was excellent.

Response: Thank you so much.

The English used in discussing the actual study was awkward, however, and should be reviewed, however. As an example, lines 272-3 "Lately a registration grid was completed recurring to the data base...." doesn't make much sense ,or 351-2 "the perceived infidelity from the victim that independently from its veracity serves as crime motive.." is unclear. While these lapses are understandable when English is a second language, good proofreading would help smooth this out.

Response: The English was improved. To clarify the procedure, the section was reformulated (see 2.3).

Reviewer 2 Report

Comments and Suggestions for Authors

Examination of the manner in which media are framing cases of domestic violence is very necessary. It is clear that the two Portuguese media outlets are framing cases very differently. The differences can have an impact on the public's thinking about domestic violence, especially when members of the public may only be reading articles from a single media outlet. The author mentions that Portugal recently changed its law to make domestic violence a public crime. It would be helpful in the discussion section to draw this out more in terms of the implications of the findings with respect to the change in the law. How might specific ways of framing media reports impact how individuals decide whether or not to report domestic violence and/or how public attitudes may be influenced by media reports and in turn influence how the public views the law.

The methodology is a good fit for the research questions examined. It might be helpful to clarify why only two newspapers were considered rather than more. 

The background information provided about media, media's involvement in framing and constructing public narratives of crime is very strong and helps position the research conducted.

The presentation of results is well-done.

Comments on the Quality of English Language

The English language in this article is of good quality. The article should be edited again for language as there are some errors that will not be picked up by spellchecking and likely not by grammar checks. For example, on line 581 "as the reddest newspaper..." should be "as the most read newspaper". A few of these types of errors are spread throughout the document. 

Author Response

Thank you for reviewing our manuscript. The suggestions were considered and included. English was revised. Please check the new version with highlights.

Authors' Response:

Examination of the manner in which media are framing cases of domestic violence is very necessary. It is clear that the two Portuguese media outlets are framing cases very differently. The differences can have an impact on the public's thinking about domestic violence, especially when members of the public may only be reading articles from a single media outlet. The author mentions that Portugal recently changed its law to make domestic violence a public crime. It would be helpful in the discussion section to draw this out more in terms of the implications of the findings with respect to the change in the law. How might specific ways of framing media reports impact how individuals decide whether or not to report domestic violence and/or how public attitudes may be influenced by media reports and in turn influence how the public views the law.

Response: Thank you for your comments. We added information on the topic in the conclusions.

The methodology is a good fit for the research questions examined. It might be helpful to clarify why only two newspapers were considered rather than more. 

Response: We clarified the selection of the two newspapers (see 267-269).

The background information provided about media, media's involvement in framing and constructing public narratives of crime is very strong and helps position the research conducted.

The presentation of results is well-done.

Response: Thank you so much.

The English language in this article is of good quality. The article should be edited again for language as there are some errors that will not be picked up by spellchecking and likely not by grammar checks. For example, on line 581 "as the reddest newspaper..." should be "as the most read newspaper". A few of these types of errors are spread throughout the document.

Response: The manuscript was edited to improve its quality.

Reviewer 3 Report

Comments and Suggestions for Authors

Review of Newspaper Headlines and Intimate Partner Femicide in Portugal

This is an interesting study about how news/media headlines about intimate partner femicide (IPF) were constructed and appeared in two print journals in Portugal. The aim of the research was to “analyze news headlines on IPF crimes, to understand which crime elements are valued and if those promote victim blaming”. The specific goal was to comprehend the connection between the most valuable criminal elements of the cases and their association with direct or indirect blaming of victims.

The literature and research review is extensive and interesting however does not highlight victim-blaming within the larger context of media representations of crime. The authors introduce several important frameworks for presenting news, e.g., episodic and thematic frames, however these are not raised again in the discussion or conclusions. This is also seen with the intersectional feminism theoretical approach used in the research. These are identified but not discussed with respect to the findings.  It would be interesting to return to both frameworks as contributing to the interpretation of the results.

The study examined 853 news headlines from two main news print journals - one sensationalist and one characterized by a robust informative scope that prioritizes bigger dimensions of news. A thematic analysis was used to identify dominant themes and categories of what type of information was reported in the headlines. While the analysis does reveal victim-blaming headlines and sub-headlines, there are several other categories of reporting also identified. While these seem to be more common than victim-blaming categories, this is not explained or summarized. If the purpose was to assess if any headlines are based on victim-blaming then this is supported however it is not clear if these are a minority/majority or if this is at all relevant.

The authors state that headlines analyses are “consistent with blaming the victims’ tactics, whether it was jealousy, despair, compassion, or an honor attack… underlining the legitimation of the crime motive and, therefore, the perpetrator severely diminished accountability.” Again this is not supported by the data that are presented in the paper and need more documentation. Only a few examples are given. The authors state that the “results presented a solid pattern of victim blaming in news headlines of IPF….” This is not evident in the way that the results are presented.

This study focuses on an important issue, that of media portrayals of violence against women and can contribute to our understanding of this with some clarification and expansion of the findings.

Comments on the Quality of English Language

Some minor editing of the manuscript is suggested. 

Author Response

Thank you for reviewing our manuscript. The suggestions were considered and included. English was revised. Please check the new version with highlights.

Authors' Response:

Review of Newspaper Headlines and Intimate Partner Femicide in Portugal

This is an interesting study about how news/media headlines about intimate partner femicide (IPF) were constructed and appeared in two print journals in Portugal. The aim of the research was to “analyze news headlines on IPF crimes, to understand which crime elements are valued and if those promote victim blaming”. The specific goal was to comprehend the connection between the most valuable criminal elements of the cases and their association with direct or indirect blaming of victims.

Response: Thank you for your comments.

The literature and research review is extensive and interesting however does not highlight victim-blaming within the larger context of media representations of crime. The authors introduce several important frameworks for presenting news, e.g., episodic and thematic frames, however these are not raised again in the discussion or conclusions. This is also seen with the intersectional feminism theoretical approach used in the research. These are identified but not discussed with respect to the findings.  It would be interesting to return to both frameworks as contributing to the interpretation of the results.

Response: The discussion and conclusions were complemented by new reflections (see, for instance, 550-553).

The study examined 853 news headlines from two main news print journals - one sensationalist and one characterized by a robust informative scope that prioritizes bigger dimensions of news. A thematic analysis was used to identify dominant themes and categories of what type of information was reported in the headlines. While the analysis does reveal victim-blaming headlines and sub-headlines, there are several other categories of reporting also identified. While these seem to be more common than victim-blaming categories, this is not explained or summarized. If the purpose was to assess if any headlines are based on victim-blaming then this is supported however it is not clear if these are a minority/majority or if this is at all relevant.

Response: The categories were better explained.

The authors state that headlines analyses are “consistent with blaming the victims’ tactics, whether it was jealousy, despair, compassion, or an honor attack… underlining the legitimation of the crime motive and, therefore, the perpetrator severely diminished accountability.” Again this is not supported by the data that are presented in the paper and need more documentation. Only a few examples are given. The authors state that the “results presented a solid pattern of victim blaming in news headlines of IPF….” This is not evident in the way that the results are presented.

Response: Further information was included to support data.

This study focuses on an important issue, that of media portrayals of violence against women and can contribute to our understanding of this with some clarification and expansion of the findings.

Response: Thank you for your comments.